# Development of InDel markers associated with leaf protein content in mulberry based on whole-genome resequencing

**Jiaqi Li** [ID]*, **Zijing Shen**, **Qin Zhang**, **Jianguo Shi** *, **Fugui Jia**, **Yumei Liu**, **Huiping Luo**, **Rou Zhang**

Yulin University, Yulin, China

☯ These authors contributed equally to this work.
* 1097243283@qq.com (JL); 378279015@qq.com (JS)

## Abstract

Mulberry (*Morus alba* L.), as a crucial dual-purpose crop for medicinal and forage applications, requires genetic improvement of leaf protein content to substantially enhance its feed value. This study utilized 46 mulberry germplasms as test materials, with 29 accessions subjected to whole-genome resequencing on the Illumina HiSeq platform. Based on the resequencing data, Indel primers were designed and screened to analyze genetic diversity and develop Indel markers tightly linked to crude protein content in mulberry leaves. The results showed that a total of 1 155 585 InDel loci were detected in the 29 accessions, with 11 401 InDels located in coding regions (0.99%). Analysis of functional annotations in extreme-phenotype germplasm revealed that 2 563 InDel-containing genes within coding sequence (CDS) regions of high-protein accessions were significantly enriched in catalytic activity and metabolic processes. Further utilizing InDel variations within CDS regions, 98 pairs of InDel primers were designed, yielding a novel linked marker (Chr1-3-204) with distinct amplification patterns between high- and low-protein accessions. Validation in 46 accessions showed 89.1% accuracy (r = 0.726, P < 0.01). Sanger sequencing confirmed its high accuracy (97.7% similarity to reference), with 84% identity to *Morus motabilis*. This study is the first to develop a linked marker for crude protein content in mulberry leaves, providing a technical reference for the breeding of high-protein mulberry varieties.N

## Introduction

Mulberry (*Morus alba* L.), a perennial woody plant in the Moraceae family, has been cultivated in China for over 4 000 years. Exhibiting robust traits such as extensive root systems, salinity-alkali tolerance, resistance to barren soils, and exceptional ecological adaptability, it is now widely cultivated across global temperate and tropical regions including Russia, Europe, Japan, and North America [1,2]. The leaves serve

**Data availability statement:** The raw sequencing data supporting the findings of this study have been deposited in the NCBI Sequence Read Archive (SRA) under the BioProject accession number PRJNA1304463. These data are publicly available and can be accessed through the NCBI SRA database. The gel electrophoresis images and other supplementary materials are available in the PLOS ONE submission system as Supporting Information.

**Funding:** This study was supported by the General Project of Shaanxi Provincial Key R&D Program (2024JC-YBQN0224), Key Laboratory Project of Scientific Research Program of Shaanxi Provincial Department of Education (23JS069), Yulin City Science and Technology Plan Project (2024-CXY-131), Doctoral Scientific Research Start-up Fund Project of Yulin University (2023GK11), Innovation Team Project for High-efficiency Cultivation and Value-added Development of Ecological Mulberry in Northern Shaanxi (2025RS-CXTD-038), Key Special Project of Shaanxi Forestry Science and Technology Innovation (SXLK2023-02-36), Project of Yulin Municipal Bureau of Science and Technology (2023-CXY-171). The funders had no role in study design, data collection and analysis, decision to publish, or preparation of the manuscript.

**Competing interests:** The authors have declared that no competing interests exist.

as a premium plant protein source, characterized not only by significantly higher crude protein content compared to conventional forages but also by their richness in all 18 amino acids and bioactive constituents including polyphenols and polysaccharides. Consequently, mulberry leaves critically determine both the nutritional value of livestock feed in China and the economic efficiency of sericulture, establishing them as a strategic resource for addressing protein feed shortages in China's animal husbandry sector [3,4]. However, current mulberry germplasm resources lack a unified classification system, and standardized naming conventions remain unestablished for the existing 200 + varieties. Traditional breeding methods suffer from prolonged cycles and low efficiency, impeding progress in mulberry cultivar development [5]. Against this backdrop, elucidating genetic variation mechanisms in mulberry and accelerating the breeding of high-protein cultivars constitute the primary objective and challenge for forage crop breeders.

The Northern Shaanxi region serves as a traditional ecologically suitable zone for mulberry in China, where its distinctive geographical environment-characterized by annual precipitation of 350−600 mm and soil pH > 8.0-has fostered unique mulberry germplasm resources. Our preliminary research revealed significant divergence in protein content (11.90%−18.68%) among 29 locally adapted mulberry varieties. This phenotypic variation not only stems from the genetic basis but also reflects the plant's adaptability to the environment (for instance, the expression level of drought-resistant LEA protein increases by 2.1 to 3.4 times [6]). Nevertheless, this valuable genetic diversity reservoir faces substantial challenges in conventional high-protein variety breeding practices: Phenotype-based selection methods are significantly compromised by environmental variability, particularly annual precipitation fluctuations and growth-stage factors (e.g., leaf position variations) [7]. Furthermore, it is urgently necessary to establish a DNA-level identification technology system for the protein content of mulberry trees in northern Shaanxi.

Whole genome resequencing, as a pivotal application of next-generation sequencing (NGS) technologies, offers exceptional advantages including high genomic coverage, superior detection sensitivity, and outstanding cost-benefit efficiency [8]. This approach has demonstrated significant success in crop genetic improvement, achieving large-scale implementation in staple crops such as rice [9], maize [10], and potato [11]. Furthermore, it has been effectively deployed in important forage species like alfalfa [12] and agropyron [13], as well as perennial woody cash crops including osmanthus fragrans [14] and populus cathayana rehder [15] for germplasm characterization, key trait gene mining, and evolutionary analyses. However, research remains scarce regarding the development of trait-linked InDel markers for mulberry and their application in germplasm identification.

DNA molecular markers have evolved as essential tools for germplasm genetic diversity research and molecular breeding, developing multi-generational technical systems including Random amplified polymorphic DNA (RAPD), Cleaved amplified polymorphic sequences (CAPS), Inter-simple sequence repeat (ISSRs), Simple sequence repeats(SSR), and

Insertion-deletion markers (InDel) [16]. Among these, InDel markers have gained extensive application in plant population genetic structure analysis, target-linked marker development, and key trait gene localization due to their significant advantages: codominant inheritance, genome-wide distribution, and high detection stability (reproducibility >98%) [17,18]. Traditional InDel marker development primarily relies on PCR amplicon sequencing alignment, which suffers from prolonged development cycles and high costs, thus failing to meet modern breeding demands [19]. With advances in high-throughput sequencing technologies, genome-wide resequencing enables scalable mining of plant InDel loci. When integrated with allele frequency association analysis, this approach efficiently screens for molecular markers linked to target traits. Nevertheless, systematic development of protein-content-linked InDel markers for mulberry leaves remains critically underdeveloped. Therefore, developing tightly linked InDel markers for leaf crude protein content traits in mulberry based on whole-genome resequencing data is both scientifically critical and urgently required.

Based on the aforementioned technical background, this study targets 29 elite mulberry varieties suitable for cultivation in Northern Shaanxi region (including characteristic germplasms such as drought-tolerant types and high-protein types), to systematically mine InDel variations through whole-genome resequencing. It further develops InDel markers significantly linked to mulberry leaf protein content traits. The expected outcomes will provide technical references for molecular assessment of leaf protein content in mulberry varieties adapted to northern Shaanxi, and establish a foundation for molecular marker-assisted breeding of high-protein mulberry cultivars, as well as for functional analysis and cloning of related genes.

## Materials and methods

### Test materials

The test materials were 46 mulberry germplasm resources (Table 1), which were planted in the experimental base of Yulin University, Shaanxi Province. From this collection, 29 elite germplasms demonstrating suitability for Northern Shaanxi cultivation were selected for whole-genome resequencing at Beijing Biomarker Technologies Co., Ltd. The remaining accessions were reserved for subsequent validation of InDel markers developed in this study.

### Determination of crude protein content in mulberry leaves

Fresh mulberry leaves were collected from 9:00–11:00 a.m. in July. For each material, three individual plants were selected, and two middle leaves were sampled from each plant, resulting in a composite sample of six mixed leaves. Sterile scissors were used during sampling to avoid cross-contamination. The collected samples were immediately transported to the laboratory, heated at 105°C for 30 minutes to inactivate enzymes, dried at 80°C to constant weight, and then pulverized. The dried powder passed through a 40-mesh sieve was used for protein content determination. Crude protein content was measured using the Kjeldahl method [20], with three replicates per sample tested and the average value calculated.

### DNA extraction and detection

Individual 0.2 g portions of tender leaves from all forty-six samples were weighed, snap-frozen in liquid nitrogen, and thoroughly pulverized into fine powder. Genomic DNA was subsequently isolated using a Plant Genomic DNA Extraction Kit (TIANGEN Biotech, Beijing). The integrity and degradation status of the extracted DNA were detected by 3% agarose gel electrophoresis at 110 V for 25 minutes. The qualification criteria were as follows: The DNA band was clear and single, with no obvious smearing or diffuse degraded bands. Subsequently, a subset of qualified genomic DNA was used for library construction and sequencing. The remaining DNA was diluted to 50 ng/μL, aliquoted and stored at −80°C to serve as templates for subsequent InDel marker PCR amplification.

**Table 1. The basic information of 46 tested materials.**

| No. | Variety | Variety type | Place of origin | No. | Variety | Variety type | Place of origin |
|---|---|---|---|---|---|---|---|
| 1 | Shin Ichinose | Cultivar | Japan | 24 | Taiyangdao | Cultivar | Heilongjiang, China |
| 2 | Baiguo mulberry | Wild | Liaoning, China | 25 | Naihan No.1 | Wild | Shananxi, China |
| 3 | Mizhi Mulberry | Wild | Shaanxi, China | 26 | Taiwan long mulberry | Cultivar | Taiwan, China |
| 4 | Liaolu No.1 | Cultivar | Shandong, China | 27 | Heizhenzhu | Cultivar | Hebei, China |
| 5 | Sang Tian No. 1 | Cultivar | Liaoning, China | 28 | Cesha No.1 | Cultivar | Xinjiang, China |
| 6 | Sweet mulberry | Cultivar | Sichuan, China | 29 | Hongguo No.1 | Cultivar | Shananxi, China |
| 7 | Baiyu mulberry | Wild | Hebei, China | 30 | Hongguo No.3 | Cultivar | Shananxi, China |
| 8 | White mulberry | Cultivar | Jilin, China | 31 | Miguo No.1 | Wild | Shananxi, China |
| 9 | Yilan mulberry | Cultivar | Heilongjiang, China | 32 | Xiongsang | Wild | Xinjiang, China |
| 10 | Guisang you 62 | Cultivar | Guangxi, China | 33 | Husang | Cultivar | Zhejiang, China |
| 11 | Hanza No.3 | Cultivar | Heilongjiang, China | 34 | Wanbaohong No.10 | Cultivar | Heilongjiang, China |
| 12 | Nongsang No.14 | Cultivar | Zhejiang, China | 35 | Hanguo No.2 | Cultivar | Shananxi, China |
| 13 | Jingsang | Cultivar | Hubei, China | 36 | Wanbaohong No.12 | Cultivar | Heilongjiang, China |
| 14 | Guisang you 12 | Cultivar | Guangxi, China | 37 | Wanbaohong No.1 | Cultivar | Heilongjiang, China |
| 15 | Autumn mulberry | Cultivar | North Korea | 38 | Fengchi No.2 | Cultivar | Jiangsu, China |
| 16 | Xuan792 | Cultivar | Shandong, China | 39 | Shaqiu | Wild | Shananxi, China |
| 17 | Wubao Sang | Cultivar | Shananxi, China | 40 | Sandaoheze | Wild | Shananxi, China |
| 18 | Yajin mulberry | Cultivar | Sichuan, China | 41 | Nonghusang | Wild | Shananxi, China |
| 19 | Poluosang | Cultivar | Hebei, China | 42 | Luza No.1 | Cultivar | Shandong, China |
| 20 | Jingbohu Mulberry | Cultivar | Heilongjiang, China | 43 | Qisi No.1 | Cultivar | Shananxi, China |
| 21 | Fengchi No.1 | Cultivar | Jiangsu, China | 44 | Zheza No.1 | Cultivar | Zhejiang, China |
| 22 | Qiangsang No.1 | Cultivar | Zhejiang, China | 45 | Yu71−1 | Cultivar | Jiangsu, China |
| 23 | Zhuyu No.1 | Cultivar | Heilongjiang, China | 46 | Jisang No.3 | Cultivar | Hebei, China |

## Indel site localization and analysis

The following verification of genomic DNA sample quality, fragmentation was performed using mechanical shearing (ultrasonication). The fragmented DNA underwent purification, end repair, 3'-end adenylation (A-tailing), and adapter ligation. Fragment size selection was then conducted via agarose gel electrophoresis, followed by PCR amplification to construct sequencing libraries. Libraries passing quality control were subjected to paired-end sequencing for 29 mulberry accessions on the Illumina HiSeq 2500 platform at Beijing Biomarker Technologies Co., Ltd. Raw reads were quality-assessed and filtered to obtain clean reads, with data reliability verified through GC content and Q30 (qualification standard: ≥ 85%) analyses. Clean reads were aligned to the white mulberry reference genome (genome size: 336.47 Mb, https://www.ncbi.nlm.nih.gov/datasets/genome/GCA_012066045.3/) using Burrows-Wheeler Aligner (BWA) software [21,22]. Alignment results were sorted and indexed using Samtools, with statistics (e.g., sequencing depth, genome coverage) calculated for each sampl. Genome Analysis Toolkit (GATK) was employed to detect InDel variants per sample. SnpEff software annotated InDel sites to determine genomic locations (e.g., exonic, intronic) and functional impacts (e.g., frameshift mutations). Variants in coding sequences (CDS) were prioritized due to their potential significant effects on gene function. Functionally annotated key InDel sites against NR [23], SwissProt [24], GO [25], KEGG [26,27], and COG [28] databases using Diamond software, thereby systematically elucidating functional differences between sequencing samples and the reference genome (white mulberry), along with their potential biological implications.

## InDel site screening and primer design

Based on whole-genome resequencing data, InDel sites meeting all the following criteria were selected: 1)Insertion/deletion length >30 bp; 2)Variant sites located in CDS regions; 3)Polymorphic in at least two sequenced germplasms. For each

chromosome (14 in total), seven InDel sites were randomly selected from CDS regions. Using Samtools, 200 bp flanking sequences (upstream and downstream) of the selected InDel sites were extracted from the reference genome sequence. Primer design was performed using Primer 5 software with parameters: 1) High specificity with functional annotation, GC content 40–60%, annealing temperature 47–60 °C; 2) PCR product size 100–400 bp; 3) Primer length 18–23 bp (both forward/reverse). Primers were named using "chromosome number + serial number" convention (e.g., chr1–1). All primers were synthesized by Sangon Biotech (Shanghai), centrifuged at 13 000 rpm for 30 s, dissolved in ddH$_2$O to 10 µmol working concentration, and stored at −20°C.

## Development and validation of protein-linked InDel markers in mulberry leaves

The PCR amplification was performed using genomic DNA templates from two high-protein and two low-protein mulberry germplasms. Indel markers demonstrating amplification specificity in extreme protein-content samples—yielding one band in high-protein samples and two bands in low-protein samples—were preliminarily identified as linked to leaf protein traits. These markers were further validated across 42 mulberry varieties (lines) with varying protein contents

PCR amplification system (20 µL): Including 1.5 µL of 10 × Buffer (containing Mg$^{2+}$), 0.5 µL of dNTPs at a concentration of 0.225 mmol/L, 0.25 µL of Easy Taq enzyme(5 U/µL), and 1 µL each of the upstream and downstream primers (10 µmol/µL). Genomic DNA template (50 ng/µL) 1 µL, ddH$_2$O 14.75 µL. PCR amplification procedure: Pre-denaturation at 95 °C for 3 minutes; Denaturation at 94 °C for 30 seconds, annealing at 58 °C for 1 minute, extension at 72 °C for 1 minute, 30 cycles; Extend at 72 °C for 10 minutes and store at 4 °C. The product was detected by 3% agarose gel electrophoresis.

## Sanger sequencing and sequence alignment of Indel markers

Genomic DNA from the high-protein germplasm *Morus alba* 'Yilan Sang' was used as the template for PCR amplification with specific primers of the Chr1-3-204 marker. After recovery and purification of the PCR products using the Omega Gel Extraction Kit, bidirectional sequencing was commissioned to Sangon Biotech (Shanghai) Co., Ltd. to obtain the full-length sequence of the Chr1-3-204 marker. Homologous sequence alignment was performed against the *Morus motabilis* reference genome (GenBank accession number: ASM41409v2) using the blastn program of the NCBI BLAST software.

## Results and analysis

### Genetic analysis of protein content traits in mulberry leaves

The leaf protein content of 46 mulberry accessions was measured (Table 2), ranging from 11.25% to 18.68% (dry weight basis) and exhibiting continuous variation. Among the samples, 22 high-protein materials (≥15%) and 24 low-protein materials (<15%) were distributed, providing optimal conditions for screening extreme phenotypes. Four extreme phenotypic materials were obtained through screening, namely Yilan Mulberry (18.68%), Xuan 792 (18.60%), Jingsang (11.90%), and GUI Sangyou 12 (11.25%). These materials will serve as key germplasm resources for the subsequent development of protein-linked markers. Statistical analysis revealed that both kurtosis and skewness absolute values were <1, conforming to a normal distribution (Fig 1). This indicates that leaf crude protein content follows quantitative genetic traits governed by polygenic control.

**Table 2. Normality test results of leaf crude protein content for 46 mulberry accessions.**

| Trait | Mean±SD | Max | Min | Median | Kurtosis | Skewness |
|---|---|---|---|---|---|---|
| Crude protein content/% | 15.26 ± 0.29 | 18.68 | 11.25 | 15 | 0.86 | 0.14 |

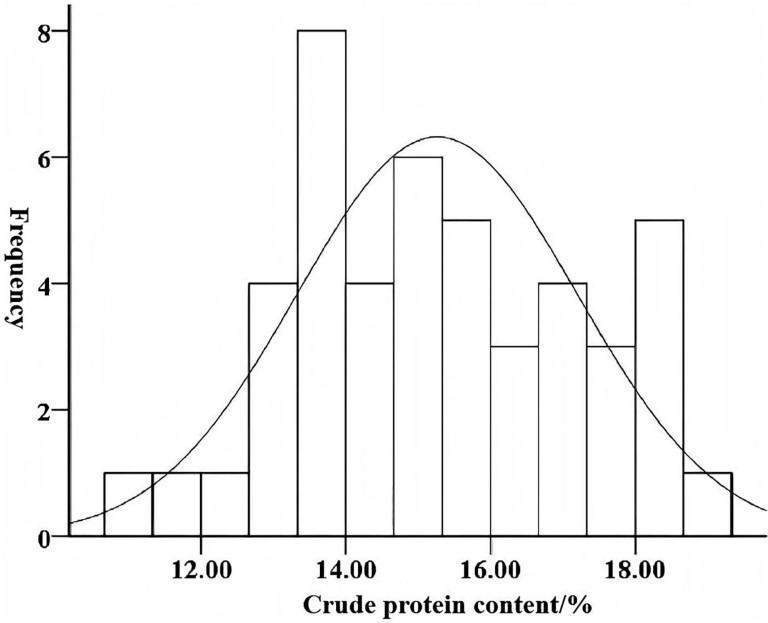

**Fig 1. Crude protein content distribution in the leaves of 46 mulberry germplasms.**

### DNA Detection

Electrophoresis analysis of genomic DNA extracted from 46 mulberry germplasm accessions revealed clear and bright DNA bands, with no protein or RNA contamination and no severe smearing. These results indicate that the quality and concentration of the extracted DNA fully meet the requirements of this study (Fig 2).

### Resequencing data analysis of mulberry

Genomic resequencing of 29 mulberry samples was performed using the Illumina HiSeq platform, yielding 584 105 114 clean reads and 174.19 Gb clean bases. The sequencing data volume per sample ranged from 4.80 Gb to 8.79 Gb, with an average of 6.01 Gb. Base quality assessment revealed that the average Q20 and Q30 values across all samples were 98.91% and 96.42%, respectively (both exceeding the 85% quality threshold). The average GC content was 36.87%, with uniform base distribution in the reads and no significant separation (Fig 3). These results indicate high sequencing base quality, which meets the requirements for subsequent bioinformatics analysis.

Clean reads post-quality control were aligned to the white mulberry reference genome (genome size: 336.47 Mb) using Burrows-Wheeler Aligner (BWA) and Samtools, achieving an average alignment rate of 98.77%, indicating robust alignment performance. Coverage depth demonstrated even distribution across chromosomes with good sequencing randomness. The average sequencing depth of the 29 samples was 16×, with individual sample depths ranging from 14× to 23×. The overall genome coverage reached 87.67%, and further breakdown of coverage metrics showed that 76.53% and 60.59% of the genome were covered at ≥5× and ≥10× depths, respectively. Collectively, these results confirm that despite an average whole-genome sequencing depth of 16×, the high alignment rate, low mismatch rate, and substantial proportion of medium-to-high depth coverage (≥10×) ensure data reliability. Additionally, these findings validate that the selected reference genome is well-annotated and information-rich, fully supporting the subsequent analytical requirements of this study.

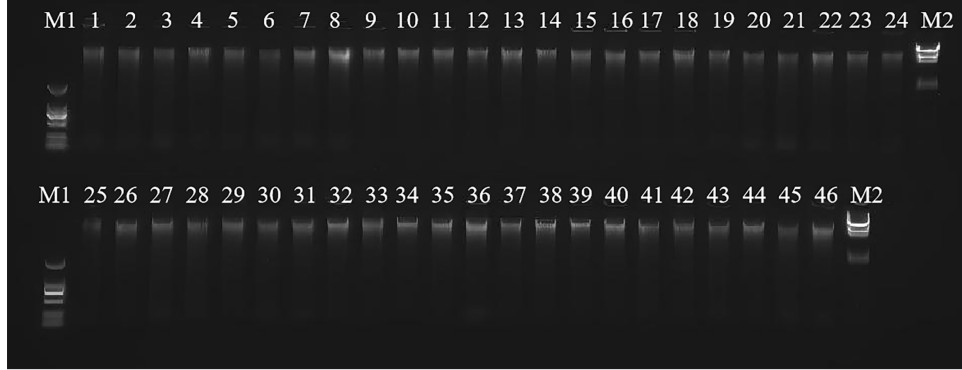

**Fig 2. Electrophoretogram of genomic DNA from 46 mulberry germplasms. Note:** M1, M2: DL2000 Marker.

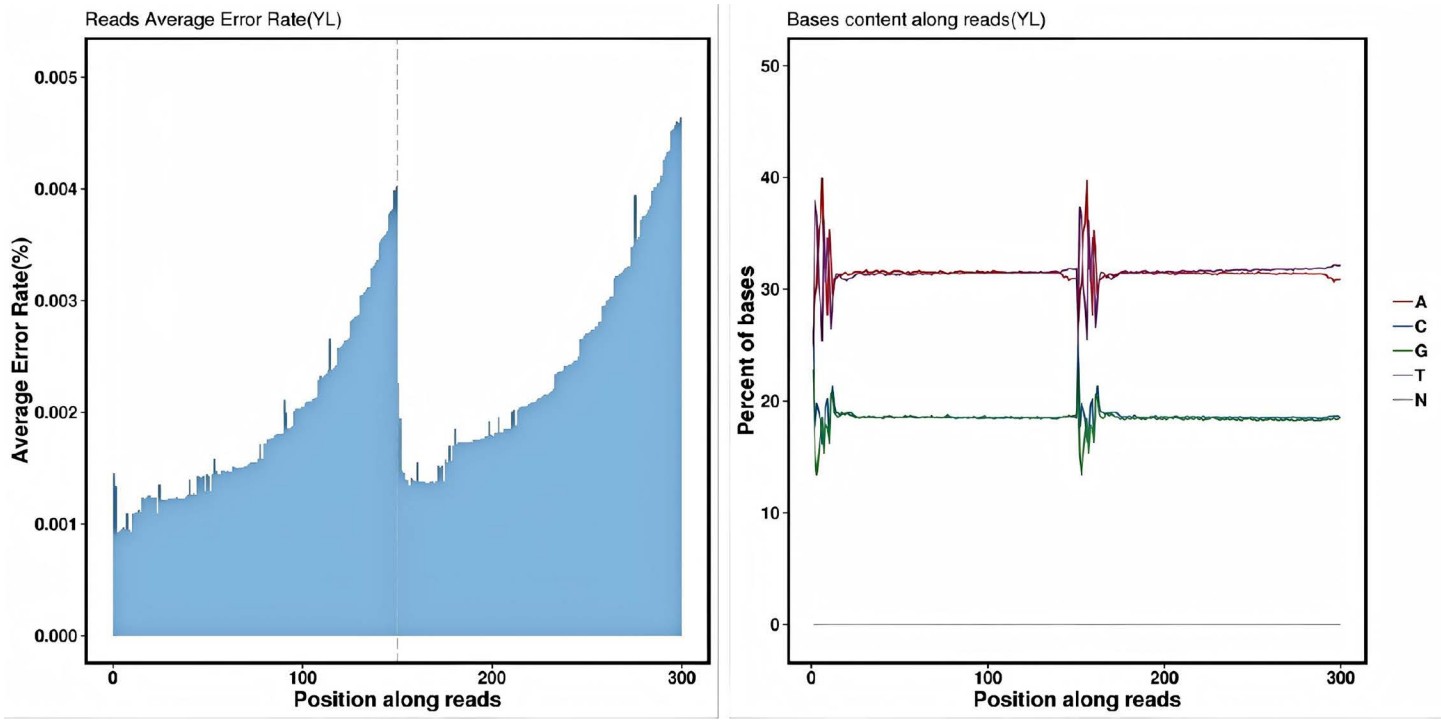

**Fig 3. Error rate of bases (left) and Clean reads base content distribution (right).**

## InDel variant analysis

Whole-genome InDel variation detection of mulberry samples was performed using GATK software. Through homology alignment with the reference genome sequence (white mulberry, genome size: 336.47 Mb), a total of 1 155 585 Indel variation sites were detected, among which 11 401 variation sites were located in the coding region (CDS). It includes 3 940 insertions and 7 461 deletions. Based on the position of the site on the reference genome and the annotation information, the occurrence region of the variant site on the reference genome, the impact and function caused by the variant site were clarified. As shown in Fig 4, a total of 1 155 584 InDel variant annotated sites were obtained. The numbers of

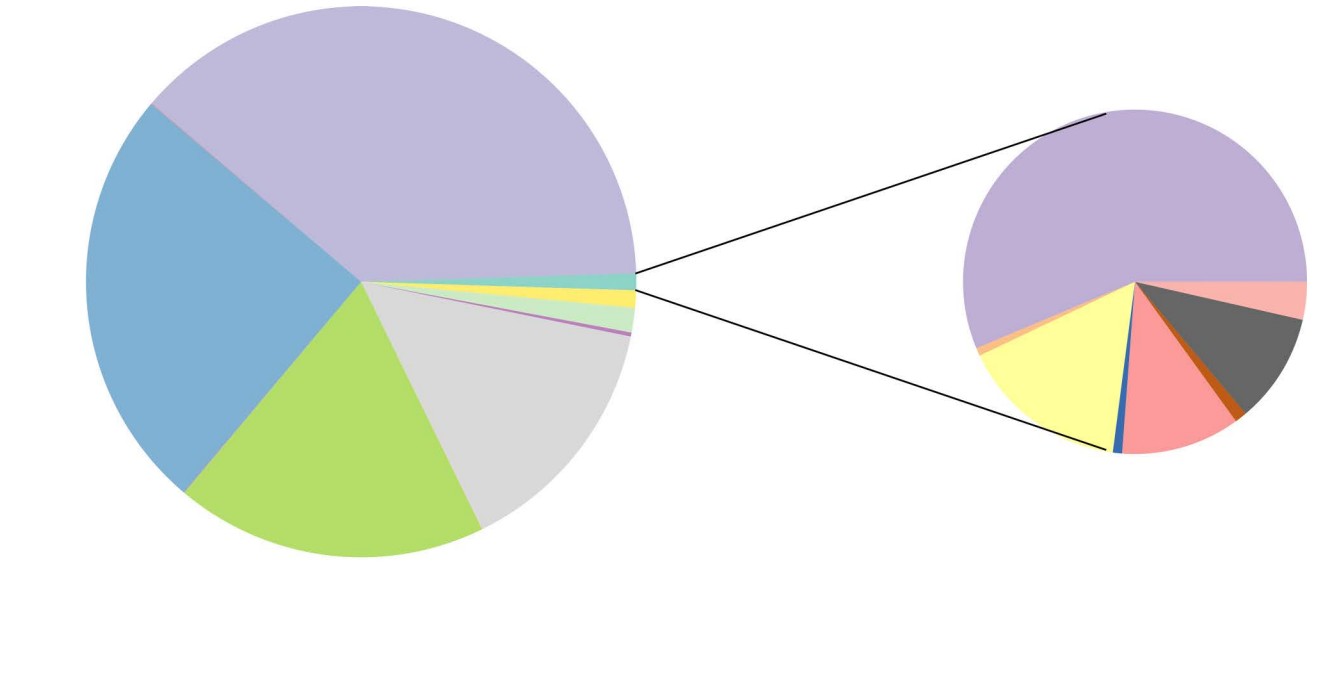

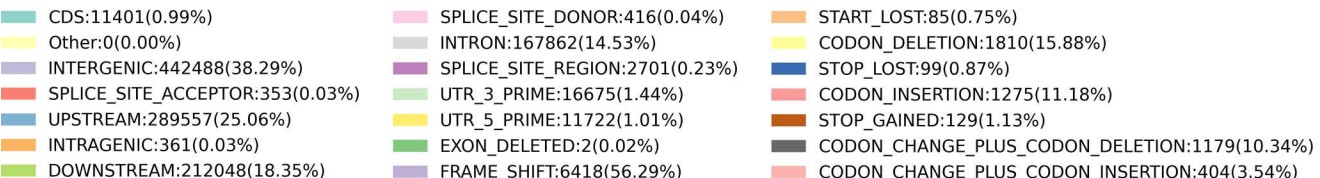

**Fig 4. Genome-wide InDel annotation classification results in mulberry samples.**

InDels in the 5'UTR and 3'UTR regions were 11 722 (1.01%) and 361 (1.44%), respectively. InDel sites located in intergenic regions and splice site regions were 442,488 (38.29%) and 353 (0.03%), respectively. Among the 11 401 InDel sites (0.98%) within the CDS region, start codon loss for 85 (0.75%), and stop codon loss for 99 (1.13%). These results demonstrate abundant InDel loci in the mulberry genome, providing critical resources for developing trait-linked InDel molecular markers and mining functional genes.

## DNA level variant gene mining

To identify key genetic markers associated with mulberry leaf protein synthesis metabolism and elucidate their genetic basis, this study focused on two extreme phenotypic varieties: high-protein Yilan mulberry (18.68% protein content) and low-protein Jingsang (11.90% protein content). Through whole-genome resequencing, functional InDel variants (including frameshift mutations in coding regions and structural variants in regulatory regions) were systematically identified, followed by differential analysis of annotated genes. There are 2 563 genes with Indel variations in the genomic coding region of high-protein Yilan mulberry, which are respectively assigned to cell composition, molecular function and biological processes. In cell composition, differentially expressed genes are mainly mapped to subclasses such as membrane, membrane part, cell part, cell, and organelle. The differentially expressed genes annotated to molecular functions are mainly concentrated in two subcategories: catalytic activity and binding, suggesting that these variant genes may be

involved in enzymatic reactions or molecular interactions and are directly related to the regulation of protein synthesis. In biological processes, differential genes are mainly mapped to two subcategories: metabolic processes and cellular processes (Fig 5). The above data indicate that the Indel variant gene of high-protein Yilan mulberry may synergistically regulate protein synthesis and storage by altering membrane structure, catalytic function and metabolic pathways, providing a molecular basis for its special phenotype. In low-protein Jingsang, 2 604 genes carrying InDel variants within genomic coding regions were functionally annotated to Gene Ontology (GO) categories: cellular components, molecular functions, and biological processes. Among cellular components, differentially expressed genes predominantly localized to three subcategories: membrane, cell part, and cell, suggesting that InDel variants may disrupt membrane integrity or organelle functionality. For molecular functions, variant-enriched genes were significantly clustered in catalytic activity and binding subcategories. Within biological processes, these genes primarily mapped to metabolic process and cellular process subcategories (Fig. 5). Collectively, this indicates that the low-protein phenotype in Jingsang likely stems from extensive impairment of cellular architecture, catalytic capacity, and metabolic networks mediated by InDel mutations in coding regions.

Based on KEGG database analysis (Fig 6) [27], significant co-enrichment (P < 0.05 and Q < 0.05) was observed in the "Other glycan degradation (KO00511)" pathway between high-protein Yilan mulberry and low-protein Jingsang varieties. Within this pathway, 63 conserved candidate genes were identified, with four predominant oxidoreductases highlighted: monooxygenase 1, squalene monooxygenase, partial squalene monooxygenase, squalene epoxidase. This finding demonstrates that despite substantial differences in protein content, both mulberry varieties maintain similar metabolic flux partitioning patterns in glycan degradation through a conserved oxidoreductase enzymatic apparatus, suggesting evolutionary convergence in this critical metabolic pathway.

## Development of Indel markers linked to mulberry leaf protein

Based on whole-genome InDel annotation of mulberry samples, 98 InDel primer pairs were designed targeting loci within coding regions that induced frameshift mutations or start codon loss, etc (S1 Table). Screening these primers using DNA from two high-protein (Yilan mulberry, Xuan 792) and two low-protein (Jingsang, Guisangyou 12) mulberry germplasms yielded 36 primer pairs amplifying clear polymorphic bands across germplasms with differential protein content.

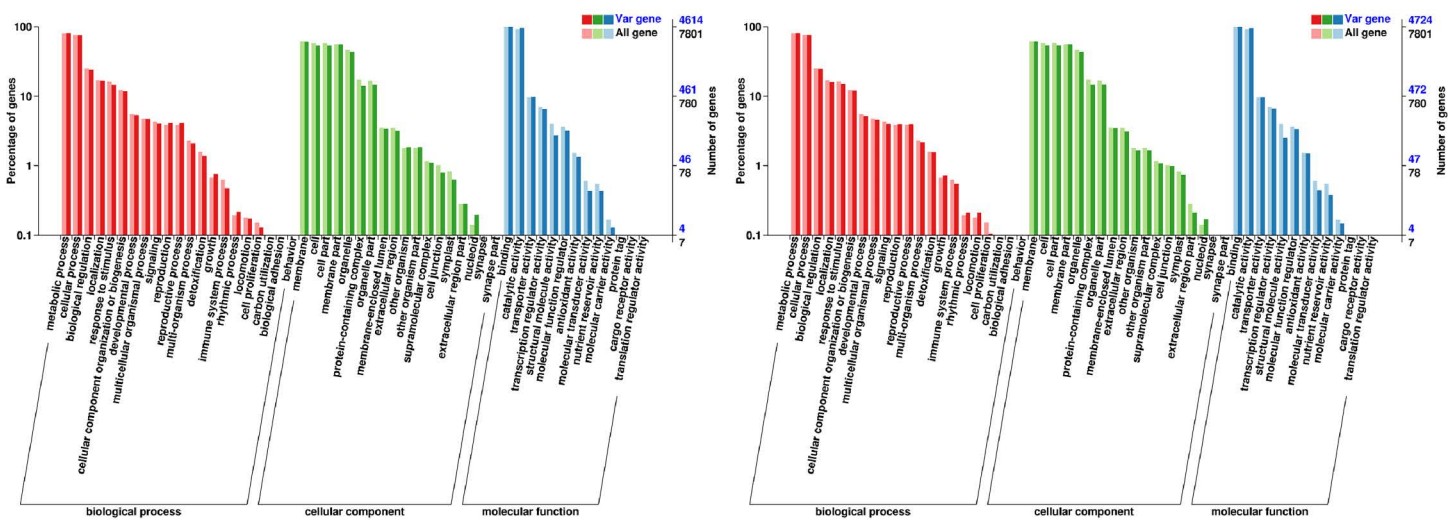

**Fig 5. GO annotation clustering of differentially expressed genes in Yilan mulberry (left) and Jingsang (right) samples.**

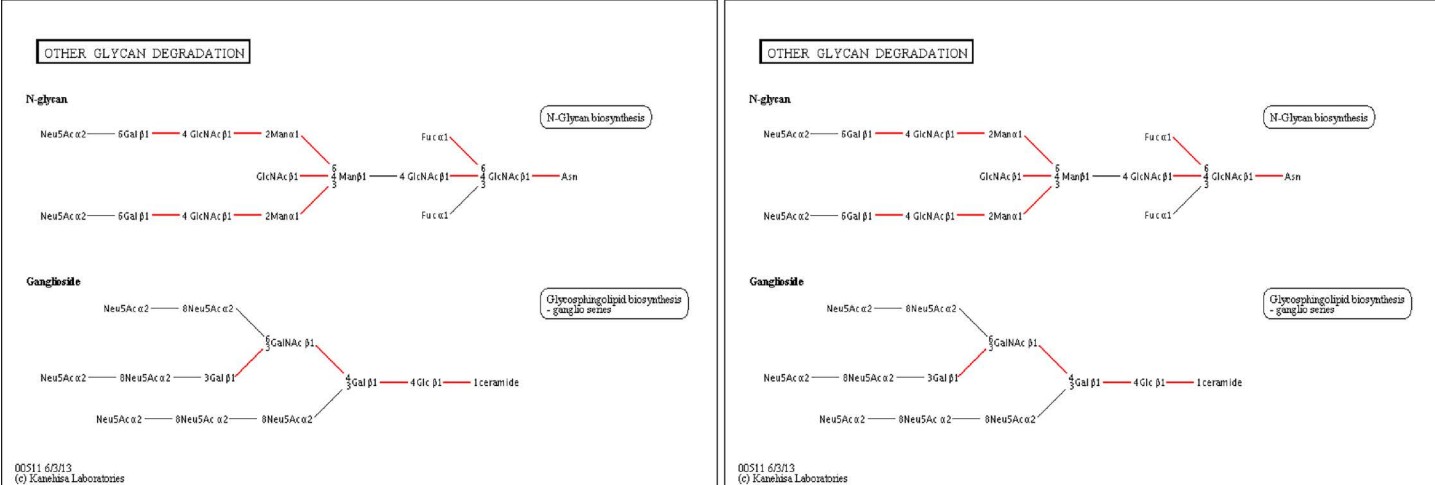

**Fig 6. Metabolic pathway mapping of differentially expressed genes in Yilan mulberry (left) and Jingsang (right) samples.**

Subsequent validation of PCR product specificity via 3% agarose gel electrophoresis facilitated the development of Chr1-3-204, an InDel marker linked to crude protein content. This marker exhibited distinct amplification patterns: a monomorphic single band in high-protein accessions versus polymorphic double bands in low-protein accessions (Fig 7).

## Validation of novel specific InDel marker

Among the 46 mulberry tree samples, 22 were high-protein (≥15%) and 24 were low-protein (<15%) materials. The 46 samples were selected to further verify the specificity of the Chr1-3-204 labeling. The validation results revealed that among the 23 accessions exhibiting a monomorphic band pattern (single amplification band), 20 were high-protein materials, demonstrating an 87% concordance rate between marker detection and phenotype; similarly, among the 23 accessions showing a polymorphic pattern (dual amplification bands), 21 possessed low protein content, achieving 91.3% phenotypic concordance with the marker genotyping results. The overall accuracy of marker-phenotype correspondence reached 89.1% across extreme protein-content accessions (Fig 8, Table 3). SPSS correlation analysis revealed a highly significant (r = 0.726, P < 0.01) association between crude protein content and marker genotypes. These results confirm Chr1-3-204 as a stable molecular marker tightly linked to mulberry protein content, enabling its application in marker-assisted breeding for high-protein mulberry varieties.

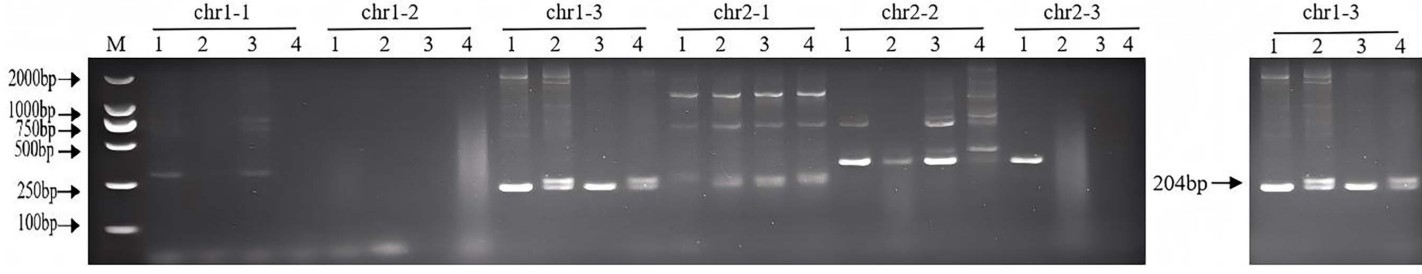

**Fig 7. The screening results of partial InDel primers across four mulberry materials. Note:** M: DL2000 Marker; 1. Yilan mulberry; 2. Jing sang; 3. Xuan 792; 4. Gui sangyou 12; chr1-1, chr1-2, chr1-3, chr2-1, chr2-2, chr2-3 represent partial InDel primer IDs. Cropped gels are shown in this figure; full original images are presented in S1_raw_images (1).

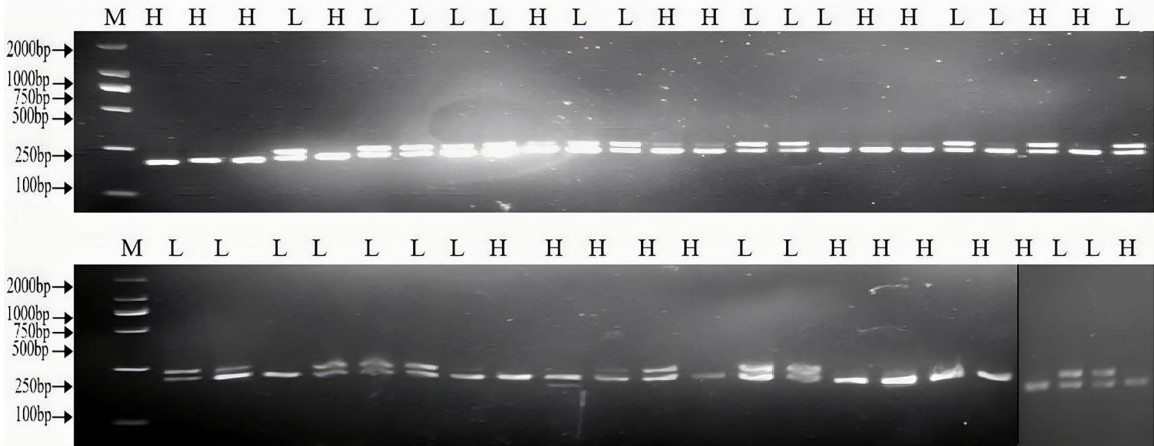

**Fig 8. Electrophoretic profile of PCR amplification with the Chr1-3-204 marker across 46 mulberry accessions.** Note: M: DL2000 Marker; "H" represents high protein content accessions; "L" represents low protein content accessions; Cropped gels are shown in this figure; full original images are presented in S1_raw_images(1). The samples derive from the same experiment and that gels were processed in parallel.

Table 3. Statistics of crude protein phenotypes and marker detection results in mulberry germplasm leaves.

| Marker | Marker test | Sample number | Phenotype | Number | Percentage/% | Total correspondence of extreme plants/% |
|---|---|---|---|---|---|---|
| Chr1-3-204 | Positive | 23 | High protein content | 2 | 91.3 | 89.1 |
| | | | Low protein content | 21 | | |
| | Negative | 23 | High protein content | 20 | 87.0 | |
| | | | Low protein content | 3 | | |

## Sanger sequencing of Indel markers and conservation analysis among bifferent species

Sanger sequencing was performed on the PCR products amplified with the specific primers of the Chr1-3-204 marker. The sequencing sequence of this marker was aligned with the reference genome (Fig 9), and the results showed that the sequence similarity between the two was 97.7% except for the Indel deletion site, indicating that the Chr1-3-204 marker had high accuracy. Furthermore, the sequencing sequence of this marker was aligned with that of another Morus species, *Morus motabilis* (GenBank accession number: ASM41409v2). A high sequence identity (84%) of the Chr1-3-204 marker was found in *Morus motabilis*, suggesting that this marker can be used for the screening of protein content traits in *M. motabilis* and its related species.

## Discussion

### The application of genomic resequencing technology in the development of Indel markers

In recent years, preliminary progress has been made in the development of insertion-deletion (InDel) markers across different species of the genus *Morus*. However, significant limitations remain, which hinder efforts to meet the research needs for in-depth genetic improvement and molecular breeding of mulberry. For instance, Rukmangada et al. [28] identified 8081 InDel loci, along with a large number of single nucleotide polymorphism (SNP) and simple sequence repeat (SSR) variant loci, by performing transcriptome sequencing on mulberry genotypes with contrasting traits and aligning the transcriptome data to the reference genome of *Morus notabilis*. Nevertheless, such transcriptome-based studies are confined to expressed gene regions and thus fail to capture structural variation information at the whole-genome scale. To address the aforementioned research limitations, the present study employed whole-genome resequencing (WGR) technology, which exhibited significant technical advantages. Whole-genome resequencing of 29 mulberry accessions identified 1 155 585

| white mulberry | ATTCTGAATCGTTCTTTATACCATGAATAGCTATTCAATTAATTCTTTGTAACCTGAATA | 60 |
| Chr1-3-204 | ATCTGAAATCGTTCTTTATACCATGAATAGCTATTCAATTAATTCTTTGTAACCTGAATA | 60 |
| | ** . ******************************************************** | |
| white mulberry | ACTATCCAAATAATTTCACAAAATCTACCTAATTTAGAAGGCCGTGATTACACCAGACAT | 120 |
| Chr1-3-204 | ACTATCCAAATAATTTCACAAAATCTACCTAA————————————————ACAT | 120 |
| | ******************************** **** | |
| white mulberry | TTAAACTTCTTTAAAATAATCACTTTTCAGAAGACTGCTTAAGCTTTACAAAATCCTCAA | 180 |
| Chr1-3-204 | TTAAACTTCTTTAAAATAATCACTTTTCAGAAGACTGCTTAAGCTTTACAAAATCCTCAA | 180 |
| | ************************************************************ | |
| white mulberry | AATTCAATGTCATGGCGA | 198 |
| Chr1-3-204 | AATTCAATGTCATGGCGA | 198 |
| | ****************** | |

**Fig 9. Sequence alignment of the gene marked by Chr1-3-204 and the reference genome sequence.**

InDel variants, with 11 401 Indel loci in coding sequences (CDS), 167 862 (14.53%) in intronic regions, and 11 722/16 675 in 5'-UTR/3'-UTR regions respectively. This important discovery is significantly contrasts with transcriptome-based results from Wang et al. [29], where only 1–3 bp small-fragment InDels were detected in the 'Anshen' cultivar, with >10 bp deletions markedly outnumbering insertion mutations. This study has not only overcome the limitations of conventional transcriptome sequencing—which is confined to expressed regions—but also systematically captured structural variations in intronic and non-coding regions through whole-genome resequencing. This breakthrough holds particular significance for mulberry research, where functional gene annotation remains notably incomplete. Furthermore, the comparison of technical methods further reveals that the targeted development strategy of Indel markers based on genomic resequencing has multiple advantages over traditional methods for developing markers: First, genome resequencing achieves significantly higher efficiency in InDel marker development compared to conventional methods. Leveraging high-throughput sequencing platforms, this technology enables single-pass whole-genome scanning, rapidly identifying millions of InDel loci to facilitate large-scale marker development [30,31]. For instance, resequencing of 29 mulberry accessions in this study detected over 1.15 million InDel sites—a stark contrast to traditional methods that rely on length-polymorphism-based amplification of gene fragments, which suffer from inherent low throughput and uneven coverage. Second, leveraging the white mulberry genome as reference, we designed 98 candidate primer pairs targeting InDel loci within coding sequences. Following rigorous screening, 36 polymorphic primers were successfully developed, achieving a 36.7% development success rate. Critically, trait association analysis revealed that marker Chr1-3-204 exhibited significant correlation with crude protein content in mulberry leaves. This strategy transcends the limitations of conventional random screening for trait-linked markers, accomplishing an efficiency leap from "blind selection" to "precision targeting". Third, genome resequencing enables simultaneous functional annotation of genes associated with InDels, thereby facilitating trait-variant linkage—a critical capacity unattainable through conventional methods that lack functional context for genetic variations. This multiplex advantage allowed our study to achieve in a single experiment the marker development scale that traditionally requires multiple rounds of screening. Notably, the resequencing-derived marker system establishes a high-density molecular toolkit for constructing genetic maps and advancing gene localization studies in mulberry.

## Development of crude protein linkage markers in mulberry leaves

Molecular marker-assisted breeding serves as a cornerstone of modern breeding technologies, with the development of markers tightly linked to target traits being its core objective [16]. However, research on molecular markers associated with key traits in mulberry remains relatively underdeveloped. Current studies have largely focused on analyzing mulberry's

population structure and genetic diversity. For example, Cai et al. [32]utilized 5 SSR primers to analyze 36 mulberry germplasms, detecting 32 genotypes and classifying the materials into three major clusters. Xie et al. [33] further corroborated this trend by employing 10 pairs of SSR primers to analyze 146 mulberry germplasms from the Jiading region in the Sichuan Basin. Their study achieved an 80.9% polymorphism rate (76/94), with population genetic analyses revealing kinship relationships among half of the cultivars. In contrast, studies on trait-linked marker development are notably limited. Shui et al. [34]successfully developed three markers (CHR2-SCAR, DFR-SCAR, and OMT1-SCAR) using 11 mulberry accessions with distinct fruit colors, among which CHR2-SCAR specifically identified yellow-green fruited varieties. Notably, the development of molecular markers tightly linked to crude protein content in mulberry leaves remains unreported to date. This discrepancy is incongruous with mulberry's status as a vital economic crop, given that leaf crude protein levels directly determine its value as both feed and functional food. Consequently, establishing such markers would not only bridge this research gap but also provide novel technological support for quality-oriented mulberry breeding.

The accuracy of marker development is a critical determinant in the success of molecular marker-assisted breeding. The Chr1-3-204 marker developed in this study exhibited high resolution (89.1% concordance rate) among 46 mulberry germplasms. However, incongruence between amplification bands and phenotypic identification persisted in 10.9% of samples, attributable to three key factors: 1) Polygenic Regulation: Crude protein content in mulberry leaves constitutes a quantitative trait governed by multiple genes, where minor-effect genes may disrupt marker-trait associations [35]; 2) Environmental Modulation: Fluctuations in cultivation conditions (temperature, precipitation, solar radiation, soil fertility) induce subtle variations in leaf protein content; 3) Technical Variability: Human error during protein quantification or electrophoretic detection impacts result reliability. These observations parallel marker-environment interaction phenomena reported in wheat [36] and seabuckthorn [37], underscoring the necessity of multi-environment phenotyping and gene editing validation to enhance marker predictive accuracy.

## Conclusions

This study investigated the genetic basis and molecular mechanisms underlying protein content in mulberry leaves through integrated analysis of phenotypic data, whole-genome resequencing, and molecular marker technologies. Through whole-genome resequencing of 29 mulberry germplasms, 1 155 585 InDel loci were identified, among which 11 401 were located in coding regions. Functional annotation analysis of extreme-phenotype germplasms—specifically Yilan mulberry (18.68% crude protein) and Jingsang (11.90% crude protein)—revealed that 2,563–2,604 genes harboring CDS-region InDel variations were significantly enriched in metabolic processes and catalytic activity-related pathways. Notably, the conserved "Other glycan degradation" pathway (KO00511), which involves oxidoreductase systems, was prominently enriched, suggesting its potential role in mediating carbon-nitrogen metabolic balance and subsequent protein accumulation in mulberry leaves. Based on the variant sites in the coding sequence (CDS), 98 pairs of candidate primers were designed, and a marker (Chr1-3-204) tightly linked to the crude protein content trait of mulberry leaves was successfully developed, with an overall accuracy rate of 89.1% (r = 0.726). Sanger sequencing confirmed the high accuracy of the marker (97.7% sequence similarity with the reference genome), and cross-species conservation analysis showed 84% sequence identity with *Morus motabilis*, indicating its potential application in related Morus species. In summary, this study reports the development of a linked InDel marker for crude protein content in mulberry leaves, providing crucial technical support for breeding high-protein mulberry varieties.

## Supporting information

**S1 Table. Base sequences of InDel primers.**
(DOC)

## Acknowledgments

We would like to thank all authors for their contributions to the study design, data collection, and manuscript preparation.

## Author contributions

**Conceptualization:** jiaqi LI.

**Data curation:** Yumei Liu, Huiping Luo.

**Investigation:** Zijing Shen, Qin Zhang, Rou Zhang.

**Methodology:** Fugui Jia.

**Writing – original draft:** jiaqi LI.

**Writing – review & editing:** Jianguo Shi.

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
