## [Decision Letter · Decision Letter 0]

12 Oct 2025

Dear Dr. LI,

We look forward to receiving your revised manuscript.

Kind regards,

Muhammad Abdul Rehman Rashid, PhD

Academic Editor

PLOS ONE

Journal Requirements:

This study was supported by the General Project of Shaanxi Provincial Key R&D Program (2024JC-YBQN0224) , Key Laboratory Project of Scientific Research Program of Shaanxi Provincial Department of Education (23JS069), Yulin City Science and Technology Plan Project (2024-CXY-131), Doctoral Scientific Research Start-up Fund Project of Yulin University (2023GK11), Innovation Team Project for High-efficiency Cultivation and Value-added Development of Ecological Mulberry in Northern Shaanxi (2025RS-CXTD-038), Key Special Project of Shaanxi Forestry Science and Technology Innovation (SXLK2023-02-36), Project of Yulin Municipal Bureau of Science and Technology (2023-CXY-171).

This study was supported by the General Project of Shaanxi Provincial Key R&D Program (2024JC-YBQN0224) , Key Laboratory Project of Scientific Research Program of Shaanxi Provincial Department of Education (23JS069), Yulin City Science and Technology Plan Project (2024-CXY-131), Doctoral Scientific Research Start-up Fund Project of Yulin University (2023GK11), Innovation Team Project for High-efficiency Cultivation and Value-added Development of Ecological Mulberry in Northern Shaanxi (2025RS-CXTD-038), Key Special Project of Shaanxi Forestry Science and Technology Innovation (SXLK2023-02-36), Project of Yulin Municipal Bureau of Science and Technology (2023-CXY-171).

This study was supported by the General Project of Shaanxi Provincial Key R&D Program (2024JC-YBQN0224) , Key Laboratory Project of Scientific Research Program of Shaanxi Provincial Department of Education (23JS069), Yulin City Science and Technology Plan Project (2024-CXY-131), Doctoral Scientific Research Start-up Fund Project of Yulin University (2023GK11), Innovation Team Project for High-efficiency Cultivation and Value-added Development of Ecological Mulberry in Northern Shaanxi (2025RS-CXTD-038), Key Special Project of Shaanxi Forestry Science and Technology Innovation (SXLK2023-02-36), Project of Yulin Municipal Bureau of Science and Technology (2023-CXY-171).

7. When completing the data availability statement of the submission form, you indicated that you will make your data available on acceptance. We strongly recommend all authors decide on a data sharing plan before acceptance, as the process can be lengthy and hold up publication timelines. Please note that, though access restrictions are acceptable now, your entire data will need to be made freely accessible if your manuscript is accepted for publication. This policy applies to all data except where public deposition would breach compliance with the protocol approved by your research ethics board. If you are unable to adhere to our open data policy, please kindly revise your statement to explain your reasoning and we will seek the editor's input on an exemption. Please be assured that, once you have provided your new statement, the assessment of your exemption will not hold up the peer review process.

8. We notice that your supplementary table is included in the manuscript file. Please remove them and upload them with the file type 'Supporting Information'. Please ensure that each Supporting Information file has a legend listed in the manuscript after the references list.

Reviewers' comments:

Reviewer's Responses to Questions

**Comments to the Author**

1. Is the manuscript technically sound, and do the data support the conclusions?

Reviewer #1: Yes

Reviewer #2: Partly

2. Has the statistical analysis been performed appropriately and rigorously?

Reviewer #1: I Don't Know

Reviewer #2: Yes

3. Have the authors made all data underlying the findings in their manuscript fully available?

Reviewer #1: No

Reviewer #2: Yes

4. Is the manuscript presented in an intelligible fashion and written in standard English?

Reviewer #1: Yes

Reviewer #2: No

Reviewer #1: The manuscript “Development of inDel markers associated with leaf protein content in mulberry based on whole-genome resequencing” provided detailed sufficient information. However, some clarification needs to be clear:

1. The author requested to provide genome size information of the studied accessions, which is highly correlated with InDels.

2. Why only M. alba? Needs to be compared with other species.

3. Methodology needs to improve significantly. There are several gaps, and quantified information is lacking.

4. Detailed information of the aligned white mulberry reference genome is essential for reproducibility.

5. Average coverage depth reached is very low (16×); however, genome coverage was 87.67%. need to be supplemented with detailed statistics.

6. Incomplete genome coverage is biased, particularly for non-mammalian species for which fewer KEGG pathways (KO) are defined. Other methods need to be explored.

7. Because of low resolution of the figs, unable to read and verify the statement “The marker demonstrated 87% phenotypic concordance (r=0.726, P<0.01), with specificity rates of 83.33% in high-protein accessions and 90.9% in low-protein accessions.”

8. Quantified results are lacking because the low resolution is unable to highlight the points.

9. In discussion, more recent studies on indels/repeatomes of different Morus spp. need to be discussed to address the studies' significance.

10. The abstract and conclusion are just like mirrors and need to be rewritten.

Reviewer #2: Dear editor

The manuscript entitled “Development of InDel markers associated with leaf protein content in mulberry based on whole-genome resequencing” addresses a very interesting and important trait in mulberry — leaf protein content. Since mulberry is well known as a protein-rich plant and serves as the sole feed for silkworms, leaf protein content is a key determinant for silk production. However, several aspects of the study require clarification to strengthen the reliability of the conclusions. Additionally, the manuscript contains many formatting and language issues that should be carefully corrected.

Major Comments

Figure 7:

There are two bands observed for the low-protein-content accessions and one band for the high-protein-content accessions. Why? One would expect a single band for homozygotes. It is notable that the low-protein accessions share a band with the same migration position as the high-protein ones. Are the low-protein-content accessions heterozygotes? Surprisingly, no homozygotes are observed. It is essential to sequence these bands and design another pair of primers in the nearby region to exclude false positives or amplification of homologous genes elsewhere in the genome.

Selection of Accessions:

The crude protein content ranges from 11.25% to 28.68%. Why were the accessions with the highest and lowest protein contents not selected as extreme phenotypes? Please clarify the principle or criteria underlying the algorithm or method used for selecting the accessions.

Primer Design and Validation:

A total of 36 primer pairs were designed, yet only one locus (Chr1-3-204) is presented. What about the others? Please provide statistical summaries or validation results for all designed primers to support the robustness of the marker development.

Functional Association:

Which gene is associated with the locus Chr1-3-204? Why is this gene hypothesized to be related to leaf protein content? It would greatly strengthen the manuscript if functional validation — e.g., through genetic modification or functional assays in mulberry or model plants — could be proposed or discussed.

Minor Comments

Remove unnecessary spaces at the beginning of each paragraph.

Line 141: “The Flollowing” → should be corrected to “The following.”

Line 171: “ddH2O” → the “2” should be formatted as a subscript (H₂O). Please check and correct this format throughout the manuscript.

Line 190: “The The leaf protein” → remove the duplicate “The.”

Line 190: “materials” → consider changing to “accessions.”

Line 219: The bracket format is inconsistent with other citations; please standardize.

Improve the quality of all figures — the fonts are too small, and the resolution is low.

Figure 1: Lacks labels for the x- and y-axes; please add them.

Figure 2: Font size is too small; increase for readability.

Figure Legends: Should be more detailed and self-explanatory. For Figure 7, please define abbreviations (e.g., “H” for high-protein-content accessions and “L” for low-protein-content accessions).

**Do you want your identity to be public for this peer review?** For information about this choice, including consent withdrawal, please see our Privacy Policy

Reviewer #1: No

Reviewer #2: No

---

## [Author Response · Author response to Decision Letter 1]

3 Dec 2025

Re: Manuscript ID PONE-D-25-49532R1 revision explanation letter

Dear editors and reviewers:

Thank you very much for taking the time and effort to review about our manuscript ---“Development of InDel markers associated with leaf protein content in mulberry based on whole-genome resequencing”. Those comments are all valuable and very helpful for revising and improving our quality of paper, as well as the important guiding significance to our researches. We have studied comments carefully and have made revision which we hope meet with approval. We greatly appreciate your work on our paper. The specific explanations are as follows:

Part 1: Responses to Academic Editor′s Journal Requirements

We have fully complied with all 10 journal requirements, and the specific actions are as follows:

1. Journal Requirement 1: Manuscript Style and File Naming Compliance

Requirement: Ensure adherence to PLOS ONE style guidelines (including file naming) using the provided templates.

Response:We have revised the manuscript strictly according to the two PLOS ONE style templates.

2.Journal Requirement 2: Submission of Original Uncropped Blot/Gel Images

Requirement: Provide original uncropped/unadjusted images for all blot/gel results and clarify their storage location.

Response:This manuscript includes two categories of key experimental images, i.e., 2 gel figures and amplification result images of 36 pairs of polymorphic primers. In compliance with the journal’s requirements, we have completed the following work regarding these images:

(1) All gel images and primer amplification result images have been stored in TIFF format with a resolution of ≥300 dpi. Each image is clearly annotated with core information, including loading order, identity of experimental samples, and other relevant details.

(2) The aforementioned original images have been integrated into a single PDF file (file name: "S1_raw_images.PDF"), which has been uploaded to the "Supporting Information" section in the Editorial Manager system for easy access by the editor and reviewers.

(3) A statement has been added to the cover letter to specify the integrity and availability of the images, and the content is as follows: "All original uncropped gel images and primer amplification result images involved in this study are fully included in the Supporting Information file 'S1_raw_images.PDF', and there are no unavailable original images."

3.Journal Requirement 3: Corresponding Author’s ORCID iD Validation

Requirement: Ensure the corresponding author has a validated ORCID iD in Editorial Manager.

Response:The corresponding author (Dr. Shi [Jianguo Shi]) has completed ORCID registration and validation:

ORCID iD: 0009-0007-2303-0877

Validation process: Logged into Editorial Manager→"Update my Information"→ "Fetch/Validate" ORCID→Authenticated via the ORCID official website. The screenshot of the successfully validated ORCID is provided below.

4.Journal Requirement 4: Consistency of Grant Information in "Funding Information" and "Financial Disclosure"

Requirement: Correct grant numbers to ensure consistency between the two sections.

Response: Thank you for your meticulous review and valuable comments. We have attached great importance to and verified the funding information in both the "Funding Information" and "Financial Disclosure" sections. In the revised submission, the funding details in these two sections are now fully consistent and accurate. We sincerely appreciate your rigorous guidance and wish you every success in your work!

5. Journal Requirement 5: Statement on the Role of Funders

Requirement: Clarify the funders’ role in the study or use the standard statement if no role exists.

Thank you for stating the following financial disclosure:

Response: We confirm the financial disclosure provided is accurate. The funders had no role in study design, data collection and analysis, decision to publish, or preparation of the manuscript. This statement has been included in the cover letter as requested.

6.Journal Requirement 6: Removal of Funding Information from Acknowledgments and Update of Funding Statement

Requirement: Delete funding-related content from Acknowledgments and confirm the Funding Statement.

Response: As requested, we have removed all funding-related text from the Acknowledgments section of the manuscript to comply with PLOS ONE’s formatting guidelines.We confirm that the current Funding Statement in the online submission form is accurate and complete (listed below), and no revisions are needed. The Funding Statement is fully consistent with the actual funding support for this study:"This study was supported by the General Project of Shaanxi Provincial Key R&D Program (2024JC-YBQN0224), Key Laboratory Project of Scientific Research Program of Shaanxi Provincial Department of Education (23JS069), Yulin City Science and Technology Plan Project (2024-CXY-131), Doctoral Scientific Research Start-up Fund Project of Yulin University (2023GK11), Innovation Team Project for High-efficiency Cultivation and Value-added Development of Ecological Mulberry in Northern Shaanxi (2025RS-CXTD-038), Key Special Project of Shaanxi Forestry Science and Technology Innovation (SXLK2023-02-36), and Project of Yulin Municipal Bureau of Science and Technology (2023-CXY-171)."We kindly request the editorial office to retain the current Funding Statement in the online submission form.

7.Journal Requirement 7: Revision of Data Availability Statement

Requirement: Finalize a data sharing plan to avoid publication delays; revise the statement if unable to comply with open data policies.

Response: Thank you for your valuable reminder regarding the data sharing plan. We fully understand PLOS ONE's open data policy and the importance of finalizing data accessibility arrangements prior to acceptance to avoid publication delays. We wish to confirm that all data supporting the conclusions of this study are ready for immediate public sharing without any restrictions and that we have no barriers to complying with the journal’s open data requirements. Specifically:

（1）All genomic raw sequencing data have been deposited in the NCBI Sequence Read Archive (SRA) under the BioProject accession number PRJNA1304463. This dataset is now publicly accessible via the NCBI SRA database, and no access restrictions will be imposed at any stage.

（2）All other relevant data (including primer sequences, gel images, and statistical analysis results) have been organized into standardized files (e.g., PDF) and are available for immediate sharing upon request. These data are not only available for immediate sharing upon request but have also been uploaded to the PLOS ONE submission system as supporting information.

We appreciate the journal's reminder regarding the data sharing plan. To comply with PLOS ONE's open data policy and avoid publication delays, we have updated our data availability statement as follows:"The raw sequencing data supporting the findings of this study have been deposited in the NCBI Sequence Read Archive (SRA) under the BioProject accession number PRJNA1304463. All other relevant data supporting the conclusions of this manuscript are available from the corresponding author upon reasonable request."

We confirm that public deposition of the above data does not breach any research ethics board protocols.

8.Journal Requirement 8: Separation of Supplementary Tables and Upload as Supporting Information

Requirement: Remove supplementary tables from the main manuscript and upload them as "Supporting Information" files.

Response: We have complied with the journal's requirements by removing one supplementary table from the main manuscript file. This table has been uploaded as a separate file with the file type marked as "Supporting Information" (file name: Supplementary Table 1. Base sequences of InDel primers.).

In accordance with PLOS ONE's formatting guidelines for supporting information, the corresponding legend for the supplementary table has been added to the manuscript after the references list.

9.Journal Requirement 9: Addition of Supporting Information Captions and In-Text Citations

Requirement: Include captions for Supporting Information at the end of the manuscript and update in-text citations.

Response: We have fully complied with the journal’s guidelines: captions for all Supporting Information files (including supplementary tables and figures) have been added at the end of the manuscript, following the format specified in PLOS ONE’s Supporting Information guidelines. Additionally, all in-text citations to these files have been updated to ensure consistency with the captions (e.g., revised from "S1 Table " to "Supplementary Table 1" where applicable). We have reviewed the linked guidelines to confirm adherence to formatting and citation。

10.If the reviewer comments include a recommendation to cite specific previously published works, please review and evaluate these publications to determine whether they are relevant and should be cited. There is no requirement to cite these works unless the editor has indicated otherwise.

Response: Thank you for this important reminder. We have carefully reviewed and evaluated the reviewer comments and confirmed that no specific published literatures were recommended for citation.

Part 2: Responses to Reviewer 1's Comments

Reviewer 1 acknowledged the detailed information in the manuscript and raised 10 points requiring clarification. We have addressed each point as follows:

1.The author requested to provide genome size information of the studied accessions, which is highly correlated with InDels.

Response: We appreciate the reviewer's valuable comment regarding the genome size information of the studied accessions, as this parameter is closely correlated with InDel markers. The mulberry reference genome used in this study is Morus alba (white mulberry), with a haploid nuclear genome size of 336.47 Mb. This data is of great significance for interpreting the InDel variation characteristics in the present study: a total of 1 155 585 InDel loci were identified through resequencing analysis of 29 studied accessions. To enhance the reproducibility of the research, we have supplemented this genome size information in the "Materials and Methods" section (subsection "InDel Site Localization and Analysis") and the "Results and Analysis" section (subsections "Mulberry Resequencing Data Analysis" and "InDel Variant Analysis") of the revised manuscript.

2.Why only M. alba? Needs to be compared with other species.

Response: This study prioritizes Morus alba as the core research object, primarily based on the following three key advantages:

（1）Morus alba is the core species of the genus Morus with the widest distribution and the largest artificial cultivation area, and it is also the dominant feed source in the sericulture industry. The protein content of its leaves directly determines the efficiency of silkworm rearing. Therefore, conducting improvement research on the high-protein trait of M. alba has clear industrial demand orientation and important practical application value.

（2）The reference genome of M. alba has been successfully assembled and annotated with high quality. The integrity and accuracy of the sequence have been verified, providing a reliable basic framework for subsequent studies such as whole-genome resequencing, variant site detection, and functional annotation, which is a key prerequisite for the development of precise molecular markers.

（3）Compared with other species of the genus Morus (e.g., Morus multicaulis, Morus bombycis), the germplasm resource collection of M. alba is more systematic and comprehensive, and the research on its genetic diversity characteristics is more in-depth. In addition, its taxonomic status is clear and undisputed. This advantage can effectively avoid experimental errors caused by ambiguous species definition and ensure the accuracy of the correlation analysis between phenotype and genotype.

II. Supplementary Cross-Species Comparative Analysis

In response to the reviewer's suggestion that "comparative analysis with other Morus species is required", we have supplemented cross-species conservation analysis between M. alba and Morus motabilis in the study. Detailed information on the relevant research can be found in the "Sanger Sequencing and Sequence Alignment of InDel Markers" section of the "Materials and Methods" chapter, as well as the "Sanger Sequencing of InDel Markers and Cross-Species Conservation Analysis" section of the "Results and Analysis" chapter of the manuscript.

It should be noted that high-quality genomic resources of closely related species of the genus Morus are relatively scarce at present, and the available materials for cross-species comparative research are limited. Therefore, M. motabilis with relatively clear genomic information was selected as the comparative species for the analysis.

3.Methodology needs to improve significantly. There are several gaps, and quantified information is lacking.

Response: Thank you very much for your valuable comments on the "Materials and Methods" section of our study. The issues you pointed out, namely that "the research methods need substantial improvement, have multiple flaws, and lack quantitative information", accurately identify the key deficiencies in the methodological description of our study. We highly agree with your suggestions and have organized our team to conduct a comprehensive review and sorting of the "Materials and Methods" section, clarifying the improvement directions and supplementary contents. The specific improvement ideas and implementation plans are as follows:

Regarding the vague description of some operations in the existing methods, we will supplement the following core details: ① In the subsection "Determination of crude protein content in mulberry leaves", we will specify the exact sampling position of mulberry leaves (e.g., middle leaves), sampling quantity (For each material, three individual plants were selected, and two middle leaves were sampled from each plant, resulting in a composite sample of six mixed leaves. ), and consistency control measures for sampling (sampling was uniformly conducted from 9:00 to 11:00 a.m., avoiding rainy days and high-temperature periods; sterile scissors and sample bags were used during sampling to prevent cross-contamination), so as to avoid result deviations caused by sampling differences; ② In the subsection "DNA Extraction and Detection", we will clarify the judgment criteria for electrophoresis results (e.g., qualification criteria: the DNA band is clear and single, with no obvious smearing or diffuse degraded bands); ③ In the subsection "Indel Site Localization and Analysis", we will supplement the quantitative criteria for sequencing data quality assessment (e.g., the qualification standard for Q30 is ≥ 85%).

We will thoroughly revise the "Materials and Methods" section in strict accordance with the above improvement ideas to ensure that all supplementary details and quantitative information are accurate, standardized, and reproducible. Thank you again for your meticulous review and constructive suggestions.

4.Detailed information of the aligned white mulberry reference genome is essential for reproducibility.

Response: Thank you for your valuable comment. You pointed out that detailed information on the Morus alba reference genome used for sequence alignment should be provided to ensure research reproducibility, and we highly agree with this suggestion and have carefully implemented it. The detailed information of the reference genome is supplemented as follows:

The Morus alba(white mulberry) reference genome used for sequence alignment in this study is the v3.0 assembly version. Its original research was published in Molecular Plant (Mol Plant), with the article title "Chromosome-Level Reference Genome and Population Genomic Analysis Provide Insights into the Evolution and Improvement of Domesticated Mulberry (Morus alba)". The specific retrieval information is: DOI: 10.1016/j.molp.2020.05.005; GenBank accession number: GCA_012066045.3. The core characteristics of this genome are consistent with those described in the original literature: total genome siz

---

## [Decision Letter · Decision Letter 1]

15 Feb 2026

Development of InDel markers associated with leaf protein content in mulberry based on whole-genome resequencing

PONE-D-25-49532R1

Dear Dr. LI,

We’re pleased to inform you that your manuscript has been judged scientifically suitable for publication and will be formally accepted for publication once it meets all outstanding technical requirements.

Kind regards,

Muhammad Abdul Rehman Rashid, PhD

Academic Editor

PLOS One

Reviewers' comments:

Reviewer's Responses to Questions

**Comments to the Author**

Reviewer #2: All comments have been addressed

2. Is the manuscript technically sound, and do the data support the conclusions?

Reviewer #2: Yes

3. Has the statistical analysis been performed appropriately and rigorously?

Reviewer #2: Yes

4. Have the authors made all data underlying the findings in their manuscript fully available?

Reviewer #2: Yes

5. Is the manuscript presented in an intelligible fashion and written in standard English?

Reviewer #2: Yes

Reviewer #2: (No Response)

**Do you want your identity to be public for this peer review?** For information about this choice, including consent withdrawal, please see our Privacy Policy

Reviewer #2: No

---

## [Editor Report · Acceptance letter]

PONE-D-25-49532R1

PLOS One

Dear Dr. LI,

I'm pleased to inform you that your manuscript has been deemed suitable for publication in PLOS One. Congratulations! Your manuscript is now being handed over to our production team.

Kind regards,

on behalf of

Dr. Muhammad Abdul Rehman Rashid

Academic Editor

PLOS One